# Engineered Nanomaterials for Improving the Nutritional Quality of Agricultural Products: A Review

**DOI:** 10.3390/nano12234219

**Published:** 2022-11-27

**Authors:** Yi Sun, Guikai Zhu, Weichen Zhao, Yaqi Jiang, Qibin Wang, Quanlong Wang, Yukui Rui, Peng Zhang, Li Gao

**Affiliations:** 1Beijing Key Laboratory of Farmland Soil Pollution Prevention and Remediation, College of Resources and Environmental Sciences, China Agricultural University, Beijing 100193, China; 2China Agricultural University Professor’s Workstation of Yuhuangmiao Town, Shanghe County, Jinan 250061, China; 3China Agricultural University Professor’s Workstation of Sunji Town, Shanghe County, Jinan 250061, China; 4School of Geography, Earth and Environmental Sciences, University of Birmingham, Edgbaston, Birmingham B15 2TT, UK; 5State Key Laboratory for Biology of Plant Disease and Insect Pests, Institute of Plant Protection, Chinese Academy of Agricultural Sciences, Beijing 100193, China

**Keywords:** nano fertilizer, inorganic nutrient, organic nutrient

## Abstract

To ensure food safety, the current agricultural development has put forward requirements for improving nutritional quality and reducing the harmful accumulation of agricultural chemicals. Nano-enabled sustainable agriculture and food security have been increasingly explored as a new research frontier. Nano-fertilizers show the potential to be more efficient than traditional fertilizers, reducing the amount used while ensuring plant uptake, supplying the inorganic nutrients needed by plants, and improving the process by which plants produce organic nutrients. Other agricultural uses of nanotechnology affect crop productivity and nutrient quality in addition to nano-fertilizers. This article will review the research progress of using nanomaterials to improve nutritional quality in recent years and point out the focus of future research.

## 1. Introduction

After the issue of food and clothing, the health, safety, and nutrition of agricultural products have become increasingly prominent, and people’s demand for health and nutrition of agricultural products has increased. This is due to the continuous improvement of the gradual enrichment of material life. Consequently, enhancing the nutritional value of agricultural products has emerged as the key to satisfying consumer demand [1,2]. The amount and quality of organic and inorganic nutrients, such as dietary fibers, carbohydrates, essential amino acids, vitamins, proteins and minerals, and other phytochemicals good for human health, have an impact on the nutritional content of farm produce [3]. The factors affecting the nutritive value of agricultural commodities are complex: the quality of seeds, climatic conditions, soil conditions, and growth conditions at various stages of growth and development will have an impact on them. Finding controllable conditions in numerous influencing aspects for adjusting and promoting is required to increase the nutritional quality of agricultural goods and obtain higher quality agricultural products.

Nanotechnology limits the material size to the level of 1~100 nm, which makes it superior to other technologies. At present, it is used in many fields, such as medical treatment [4], construction [5], biomaterials [6,7], textile [8], environmental protection [9], food processing and packaging, agriculture, which brings new opportunities and challenges to traditional industry technology. Nanotechnology is placed with high hopes to achieve sustainable green development. With the development of nanotechnology and the wide application of nanomaterials, the emergence of nano-agricultural products such as nano-fertilizers, nano-pesticides, and nano-biosensors has improved the problems in traditional agriculture and improved the nutritional quality of crops. For example, mesoporous silica nanoparticles (MSNPs) can improve the nutrient delivery speed and growth speed of *Z. japonica* [10]. As the carrier of herbicide, nano-capsules can reduce the amount of herbicide without reducing the efficiency. For instance, poly (*epsilon-caprolactone*) nano-capsules can be used as herbicide carriers of atrazine [11]. Nano fertilizer can decrease the spillage of nutrients, minimize the waste of fertilizer, and promote merging the damp and alimentation [12,13]. At present, the research on the impact of nanomaterials on crops is more about nanomaterials reducing the amount of chemical agricultural products and improving their utilization efficiency, which is better than traditional agricultural products. However, there is a lack of research on the effect of nanomaterials on the nutritional quality of agricultural products, especially the improvement of inorganic nutrition and organic nutrition. This paper summarizes the factors affecting the nutritional quality of agricultural products and explores the relationship between nanomaterials and the factors affecting the nutritional quality of agricultural products. Focusing on the improvement effect of nano-fertilizers on inorganic and organic nutrition of crops, nano-pesticides enhance the efficacy by controlling release to reduce the threat of pests and diseases and ensure the nutritional quality of agricultural products, while paying attention to the phytotoxicity of nanomaterials.

## 2. Factors Influencing the Nutritive Value of Farm Commodities

The way that crops absorb and use nutrients has an impact on the nutritional value of farm produce. The nutrient absorption of the plant is mainly through root absorption and leaf absorption. The root mainly draws nutrients from the soil, and the aboveground part mainly absorbs and exchanges nutrients with the external environment through the leaf surface. Plant nutrition can be divided into organic nutrition and inorganic nutrition. Organic nutrition is closely related to photosynthesis, such as sugar, protein, organic acid, carbohydrate, and fat. Inorganic nutrition mainly water and inorganic salts.

In aspects of crop propagation and development, there are 17 kinds of necessary nutrients. The massive elements are O, C, N, H, P, K, Mg, Ca, S, and Si. The essential trace elements are Cu, Zn, B, Fe, Mo, Mn, Cl, Na, and Ni. Plant growth shows abnormal plant morphology and leaf color, poor development, poor stress resistance, and decreased fruit quality, which are related to the lack or excess of essential elements.

The nutritional quality of agricultural products on the one hand depends on the quality of seeds, on the other hand, depends on external growth conditions. For example, light affects photosynthesis, thereby affecting nutrient production and transport mechanisms; soil conditions affect the root absorption of essential nutrients; inorganic salts affect plant growth and the nutritional quality of agricultural products.

### 2.1. Intrinsic Cause: Seed Quality

Numerous genes regulate the size, composition, and quality of seed structure, which has a substantial impact on the quality of agricultural products. Therefore, choosing high-quality seeds for high-quality agricultural goods is vital to improving the nutritional content of agricultural products [14]. Recent investigations have shown that the nutritional quality of rice is mostly determined by the size of the seed and the amount, type, and physicochemical characteristics of the protein and starch in the endosperm. In addition, other substances contained in different parts of rice seeds, such as lipids, minerals, vitamins, and phytochemicals, also have an impact on the nutritional value of rice [15]. In addition, the moisture content at the optimal harvest time of most cereals depends on the type, cultivar, and seed quality [16]. The variety of legumes directly affects the blackening and hardening of seeds, and the blackening of seed coats after harvest depends on the genotype of legumes [17]. Irakli et al. found that the content of oil, protein, and carbohydrate and the composition of fatty acids in cannabis seeds were mainly affected by genotype [18]. The structure, size, and genes of seeds directly affect or determine the nutritional quality of crops. Therefore, by enhancing the structure and form of seeds as well as gene expression, we can enhance the seed quality and thus the nutritional content of agricultural goods.

### 2.2. External Factors: Growth Environment

Murube et al. analyzed the nutritional traits of 25 common legumes in different regions of Europe and found that the differences were determined by genotype, but not all the traits were inherited, which indicates that environmental conditions have a relevant impact on these nutritional compounds [19]. As a result, in addition to seed quality, other growth factors such as light, humidity, temp, soil fertility, pollution, and the use of pesticides and fertilizers will affect crop growth and development, which will then affect the nutritional value of agricultural goods.

#### 2.2.1. Climatic Conditions

Crop growth and development are influenced by climate conditions, which also have an impact on how well-balanced the diet of crops is. Huang et al. showed that the growth of leaves of two-leaf Chinese cabbage irradiated alternately with red and blue light was better than that of control treatment, moreover, red and blue light irradiation had an effect on the concentration of chla, total chlorophyll, carotenoids, soluble protein and so on [20]. In addition, Anza et al. showed that the season had an effect on the nutritional quality parameters of tomatoes, and the nutritional quality of tomatoes grown in spring was better [21]. Song et al. determined that the interplay between nutritional solution content and light intensity (NSC) had a significant impact on nutritional quality, and mineral content by investigating how the relationship between light intensity and nutrient solution concentration affects lettuce’s nutritional value [22]. Factors like temperature, light, season, and other climatic circumstances will affect how nutrient-dense agricultural goods are.

#### 2.2.2. Soil Conditions

Soil is the foundation of crop growth and the main way to absorb water and nutrition. The content of nutrients that can be absorbed and utilized in soil, soil fertility, soil moisture content, and soil pollution will affect the growth and development of agricultural products and their nutritional quality. It can be said that the quality of soil directly determines the quality of agricultural products. Li et al. show that nutrient concentration in plants is related to soil type [23]. Sanchez-Navarro et al. have proved that mineral fertilizers and organic fertilizers can help improve soil fertility and yield. Both of them can increase soil characteristics and improve the yield and nutritional quality of cowpea [24]. The quality of food crops is also significantly influenced by other agronomic factors, including the type of tillage and rotation, soil moisture, crop seed incubation, and plant genetic improvement programs, according to studies [25]. The research of Leskovar et al. shows that the irritation of onions varies with varieties, soil types, soil moisture, and other growth conditions [26,27]. Using carrots as test subjects, Ma et al. concluded that long-term usage of contaminated water could influence crop growth, lower the nutritional content of vegetable products, and worsen pollution [28]. Recently, studies have shown that saline-alkali conditions greatly changed the proteome and amino acid spectrum of quinoa seeds, thus affecting their nutritional quality [29]. The quality of soil conditions has a great impact on the yield and quality of agricultural products, and also has a certain impact on their nutritional quality.

#### 2.2.3. Growth Nutrition Conditions

To ensure the nutritional quality of agricultural products, in addition to suitable climatic and soil conditions, good growth conditions are also required. Whether or not there is insect disease, the agricultural method of farming rotation, the application of mycotoxins and pesticides, and fertilizers will affect the growth and development of agricultural products [30]. Additionally, the outcomes demonstrated that lettuce and mung bean pod growth, yield, and nutritional quality may all be enhanced by *arbuscular mycorrhizal fungi* (AMF) [31,32]. Iqbal et al. proved that forage sorghum legume intercropping system and tea tree (*Camellia sinensis L.*) plantation chestnut (*chestnut kernel*) intercropping system in temperate regions of China can greatly improve resource utilization efficiency and crop nutritional quality in agroforestry ecosystems, and achieve sustainable development of green agriculture [33,34]. El-Gioushy et al. used Washington navel orange trees as the research object and sprayed ZnSO_4_ (0 mg/L, 300 mg/L, 600 mg/L) solution on the leaves. The results showed that ZnSO_4_ could improve the growth, nutritional quality, and productivity of Washington navel orange trees [35]. Recent research has demonstrated that the use of *mepiquat chloride* (MC) can significantly boost the nutrient content and biological accumulation in seeds under the control circumstances of boron deficiency and sufficient boron, boosting the yield and nutritional quality of seed cotton [36]. Additionally, the productivity and quality of tomatoes can be increased by combining biochar with both organic and inorganic fertilizers [37]. Ronga et al., however, demonstrate that an excessive N input will negatively impact the yield and quality of crop plants planted in conditions of high soil fertility and will also lessen the sustainability of agriculture [38]. Kolencik et al. improved the yield and quality of lentil seeds with the low concentration of nano ZnO NP concentrations in foliar spray [39]. Qu et al. used the method of applying nitrogen three times and a 45% nitrogen rate to treat vegetables and obtained higher vegetable yield and quality [40]. Ronga et al., however, demonstrate that an excessive N input will negatively impact the yield and quality of crop plants planted in conditions of high soil fertility and will also lessen the sustainability of agriculture.

In short, the nutritional quality of agricultural products is mainly determined by the internal genes and genetic information of seeds. Improving seed shape, gene coding, and genetic information can improve the nutritional quality of agricultural products. Additionally, the nutritional content of agricultural products is significantly influenced by the climatic conditions, soil conditions, and growth circumstances that exist throughout the growth of agricultural products. The nutritional value of agricultural products can be increased by ensuring that they are grown under conditions of proper temperature and humidity, healthy soil, acceptable farming practices, and appropriate fertilizer conditions (Figure 1).

## 3. Effect of Nanomaterials (NMs) on Nutritional Quality of Agricultural Products

With the continuous development of nanotechnology, nanomaterials are more and more widely used in agriculture. Nano-tools have transformed traditional farming methods into precision farming. The uncontrolled use of pesticides and fertilizers in traditional agricultural production has increased food production, but soil fertility has declined dramatically. Of the traditional fertilizers and pesticides applied, 50–70% are not used due to leaching, mineralization, and biotransformation [41,42]. Their residues in organisms and their migration and transformation in the environment affect human health and cause environmental pollution. The emergence of nanomaterials, especially nano-fertilizers and nano-pesticides, has improved this situation, not only improving the utilization rate of agricultural chemicals but also improving the nutritional quality of agricultural products by improving the nutritional conditions, soil conditions and protecting crops from pests and diseases during crop growth [43]. The following will focus on the improvement of inorganic and organic nutrition of agricultural products by nano-fertilizers and the improvement of agricultural products and their nutritional quality by the controllable release of nano-pesticides.

### 3.1. Synthesis and Mechanism of Nano–Fertilizer

Nano-fertilizers include: NMs that can provide one or more nutrients to plants and promote their growth and yield, or NMs that can improve the performance of traditional fertilizers but do not directly provide nutrients to crops. The former refers to the processing of fertilizers to nanometer size. When the particle size of nanoparticles is smaller than the cell wall pores (5–20 nm), nanoparticles may enter plant cells directly through the mesh cell wall structure [44]. The latter refers to the use of engineering nanomaterials as fertilizer carriers to achieve targeted transport or release control of fertilizers. The synthesis of nanomaterials is divided into physical, chemical, and biological methods. The physical process known as “top-down” shrinks bulk materials to the nanoscale by ball-milling the materials. This method has great limitations, and the obtained particles contain more impurities. The bottom-up method relies on chemical reaction and chemical-controlled synthesis, which can better control the particle size and the number of impurities [45,46,47]. The biological synthesis of nanomaterials mainly uses plants, fungi, and bacteria. Its advantage is that it can better control the toxicity and size of particles [48]. Ekanayake et al. synthesized ZnO nanoparticles as nano-fertilizers by chemical precipitation using an aqueous solution of 0.1 mol dm^−3^ Zn(NO_3_)_2_·6H_2_O ZnO_(s)_ and 0.8 M NaOH aqueous solution [49]. Le et al. prepared NPK-hydroxyapatite nano-hybrid structure by chemical method and synthesized Ag, Fe, Cu, Co, Zn, and other trace element solutions in the form of nanoparticles by chemical reduction with NaBH_4_ as a reducing agent. Finally, the two were integrated into water-holding materials such as alginate [50]. Ha et al. prepared chitosan nanoparticles by modified ionic method and added 0.3% KNO_3_ into chitosan nanoparticle emulsion to prepare nano-fertilizer [51]. In addition, in terms of biosynthesis, Kaur et al. used bacteria to synthesize silver nanoparticles. The strain was inoculated in a medium containing 10 mmol L^−1^ AgNO_3_, and the color of the medium was observed. Brown indicates the formation of silver nanoparticles. It improved the interaction of chickpea soil flora [52]. The type, dosage, application method, and structure of nano fertilizers determine the absorption, transport, transformation, and accumulation of nanoparticles by crop roots or leaves [53], and then affect their nutritional quality [54]. The dispersion, aggregation, bioavailability, absorption, and transport mechanisms of nano fertilizers vary with the transport mechanism within plants [55,56] and are also affected by pH value, structure and texture, mineral content, soil organic matter, and microbial population [57,58]. Nano fertilizer is transported to the aboveground part through root application or foliar application, through the root epidermis and endodermis into the duct, or absorbed by the leaf pores and transported through the phloem [53,59]. Both pathways require nano fertilizers to pass through the cell wall, so only nanoparticles smaller than 8nm can reach the plasma membrane through the pores [60]. The study found that iron oxide nanoparticles (Fe_2_O_3_ NPs) might replace conventional fertilizers and enhance the nutritional value and growth of peanut plants [61]. Some studies have also shown that one kind of nanoparticle will have different effects on different parts of the same crop [62].

**Table 1 nanomaterials-12-04219-t001:** Nanomaterials as plants supplement the infinite nutrients and enhance plant growth and development.

Nutrient	Nanomaterials and Crops	Experimental Conditions	Findings	References
N	Clinoptilolite-NH_4_ Clinoptilolite-urea, *L. multiflorum*	Treated with 0, 60, 120, and 180 kg/ha of nitrogen on sandy loam soil.	Both yield and nitrogen uptake efficiency significantly improved.	[63]
	Nano nitrogen (n-N), Lettuce plant	Surface irrigation and drip irrigation are, A combination of nano nitrogen(n-N) and bulk size nitrogen (b-N).	The combination of 75% n-N drip irrigation and 25% n-N foliar spraying significantly affected plant biomass, crude protein, and yield, and improved nitrogen absorption and utilization efficiency.	[64]
P	Synthetic apatite nanoparticles, Soybean (*Glycine max*)	Fertilizing schemes: tap water, artificial fertilizer with standard P, artificial fertilizer without P, and artificial fertilizer with NHA.	Synthetic apatite nanoparticles enhanced the growth rate of soybean and the seed yield by 20.4%. Aboveground and underground biomass increased by 18.2% and 41.2%, respectively.	[65]
	synthetic nano-hydroxyapatite (NHA), lettuce (*Lactuca sativa L.*) plant	Natural light greenhouse. 200 mg of P and 200 mg of basal N (from NH_4_NO_3_) per kg of soil.	NHA can increase the dry weight of lettuce plants more than H_3_PO_4_-P (soluble phosphorus) and is more effective for the growth of lettuce plants.	[66]
K	Potassium nano-silica (PNS). Maize	PNS concentrations at 0, 100, and 200 ppm, together with irrigation (−0.03, −0.6, and −1.2 MPa) Foliar spraying	Applying PNS improved the number of inorganic nutrients in seeds under drought stress. The harmful consequences of drought stress were lessened by PNS.	[67]
Ca	Calcium carbonate nanoparticles. Thankan(*Citrus tankan Hayata*) plants.	Field tests. 95%WP, 26%SC, and CK.	The calcium concentration in leaves treated with nano-Ca increased 13 times compared to the control group, although excessive Ca spraying may impact the potassium content.	[68]
Mg	Magnesium particles are less than 900 nm. peaches (*Prunus Persica*).	Foliar fertilization. Spray 500 kg/ha diluted magnesium on 10 trees per treatment using an automatic sprayer.	The Mg content in the petiole, front, back, and leaf side of peach increased after treatment when the particle size of the plant nutrients was reduced to less than 900 nm.	[69]
Fe	Fe(III)-aminolevulinic acid nano chelate. *Portulaca oleracea L.*	foliar application, N-Fe (ALA)_3_ and Fe-EDDHA were sprayed with Fe at a rate of 0.1% and 0.2% (w/v).	Foliar application of Fe (III) -aminolevulinic acid nano-chelates can increase the content of Fe, Zn, N, Mg, Ca, and K in the aboveground part, and can increase the growth rate of purslane plants.	[70]
	nZVI, Wheat crops	Foliar application, Different iron sources (FeSO_4_, Fe- EDDHA, and nZVI, 0.2%)	N-ZVI and urea significantly increased the iron concentration in grains. After n-ZVI is likely to be used as an alternative iron source.	[71]
Mn	Manganese Nanoparticle. Wheat (*Triticum aestivum L.*).	Soil cultivation and foliar spraying.	The application of nano-manganese to soil decreased the contents of manganese, phosphorus, and potassium in plants, and the exposure of leaves increased the contents of manganese in branches and grains.	[72]
Zn	Zinc Oxide Nano Particles. Maize (*Zea mays L.*) Plant.	n-ZnO particles in suspension form (0, 0.05 ppm,0.5 ppm), set the same concentration of ZnSO_4_ as the control group.	Application of nano zinc increased plant dry weight. Branch Zn content increased to 37 ppm, root length increased by 1.6 times compared with the control, and plant height increased to 59.8 cm.	[73]
	Nanoscale zinc oxide (ZnO). Peanut.	ZnO suspensions were prepared at concentrations of 400, 1000, and 2000 ppm.	ZnO nanoparticles can transfer Zn to plants, which can improve seed germination, root length, branch dry weight, and pod yield, while too high a concentration will limit plant growth.	[74]
Cu	25 nm or 60–80 nm nano-Cu. Cowpea.	Plants were exposed to four levels (0, 125, 500, and 1000 mg/kg) of nano-Cu for 65 days.	The absorption effect of copper at the nano-copper level was more significant than that of the control group.	[75]

### 3.2. NF for Improving Inorganic Nutrition

With the continuous application and development of nanotechnology in agriculture, the advantages of nanoparticles s over traditional fertilizers have gradually become prominent, with the advantages of high absorption efficiency, controllable application time, and less nutrient loss. Reasonable application of nanoparticles can add to the absorption of essential nutrients including N, K, P, Mg, Ca, Mn, Fe, Zn, Cu, and Mo by plants, improve the inorganic nutritional quality of plants, and enhance economic and environmental benefits [76].

#### 3.2.1. N

The main forms of nitrogen absorbed by plants are NO_3_^−^-N and NH_4_^+^-N. It can also absorb NO_2_^−^-N, which can be used for the synthesis of amino acids, biocatalytic enzymes, proteins, etc. It also contributes significantly to plant organic structure compounds and living substances and is a crucial component of chloroplasts, which are where most photosynthesis occurs. Studies have shown that compared with traditional nitrogen fertilizer, nano-nitrogen fertilizer can indeed improve the utilization efficiency of N by plants. In the comparative experiments of two different application methods of surface irrigation and drip irrigation, it is found that the application method has little effect on the effect of nano-nitrogen fertilizer, which can supply more nitrogen to plants in less quantity and increase the biomass of plants. Nano-nitrogen fertilizer still performs well even under drought stress [63,64].

#### 3.2.2. P

Plant development and metabolism depend on phosphorus. It is a component of nucleic acid, nucleoprotein, phospholipid, phytin, ATP, and some enzymes. It can strengthen the synthesis and transportation of carbohydrates in plants, promote the metabolism of nitrogen and the synthesis of fat, and contribute to the increment of plant resistance. Liu et al. concluded that the synthetic apatite nanoparticles added the growth rate of soybean and the seed yield compared with the soybean treated with ordinary phosphate fertilizer (Ca(H_2_PO_4_)_2_). Aboveground and underground biomass increased by 18.2% and 41.2%, respectively [65]. Taskin et al. also confirmed that NHA can increase the dry weight of lettuce plants more than H_3_PO_4_-P [66]. Following studies on lettuce in high and low calcium soil and soybean in a peat perlite combination, it was shown that NHA may substitute conventional phosphorus fertilizer and improve crops’ nutrient quality by enhancing P uptake and utilization.

#### 3.2.3. K

One of the key macronutrients for plants, potassium is crucial for the processes of photosynthesis, respiration, and enzymatic response. It can control stomatal opening, achieve CO_2_ uptake and stabilization, and improve plants’ resilience to environmental stressors including drought stress [67].

#### 3.2.4. Ca

Plants take up calcium in the form of Ca^2+^, a structural component of the cell wall and an essential element for cell division. Especially for legume crops, higher concentrations of calcium nutrition are conducive to the formation of nodules and symbiotic nitrogen fixation. Compared with traditional fertilizers, nano-calcium fertilizer can more efficiently increase the calcium content in plants [67], thus affecting the nutritional quality of plants.

#### 3.2.5. Mg

Mg^2+^ is the most common type of magnesium taken by plants, which is an indispensable element of chlorophyll and affects the morphology of plant cell walls together with calcium. Magnesium promotes plant photosynthesis by promoting chlorophyll absorption of light energy and participates in carbohydrate metabolism, and biosynthesis of phospholipids, nucleic acids, and other compounds. Nanoscale plant nutrient magnesium is more easily absorbed by plants, a study suggests [69].

#### 3.2.6. Fe

Plant roots primarily take up iron in the form of Fe^2+^ and chelated iron, and the absorption of Fe^3+^ is very small. Iron is an important component of ferredoxin in plants. It can bind to chloroplasts and affect plant photosynthesis and a series of redox processes through electron transfer. According to studies, nano-iron fertilizers can boost plants’ iron levels and perhaps take the place of other iron sources [70,71].

#### 3.2.7. Mn

Mn^2+^, which is the form of manganese that plant roots absorb. Mn has an impact on plant respiration and photosynthesis, contributes to the conversion of NO_3_^−^ to NH_4_^+^, impacts the uptake and usage of N, and is crucial for the redox reaction in plants. Compared with traditional fertilizers, nanoparticle nutrients are more easily absorbed and utilized by plants and have an impact on plant growth, yield, elongation of branches and roots, and the content of nutrients in each part. However, different application methods have different effects. The effect of applying nano-manganese in the soil is not as good as that of spraying nano-manganese on leaves [72].

#### 3.2.8. Zn

In plants, zinc is primarily taken in the form of Zn^2+^, which is essential for auxin production and growth. Compared with traditional zinc fertilizer (ZnSO_4_), nano-sized ZnO can better promote the growth of plant dry weight, branches, root length, and height, and promote plant growth [73]. In addition, some scholars have carried out experiments and discussions on more effective nano-sized ZnO concentrations for different plants [74,77].

#### 3.2.9. Cu

Copper plays a role in the respiration, metabolism, and photosynthesis of plants and is absorbed by plant roots as either Cu^2+^ or Cu^+^. The nano-copper treatment of cowpea can increase the absorption and bioaccumulation of copper. The highest copper content of nanomaterials with different sizes is different. The copper accumulation of nano-copper particles (<25 nm and 60–80 nm) increases with the increase in treatment level. At the same time, nano-copper exposure can also produce toxicity and lead to oxidative stress in cowpea [78]. Therefore, it is necessary to control the conditions that may affect crop growth, such as particle size and application method [75].

In comparison to conventional fertilizers, studies (Table 1) have demonstrated that nanomaterials can more effectively promote nutrient uptake and use by plants. Compared with soil cultivation, the foliar spraying method of crop absorption of nutrients is more significant. The size and concentration of nanoparticles will affect the effect of nano-fertilizers, and may also produce toxic reactions. The effects need to be further studied [79,80].

### 3.3. NF for Improving Organic Nutrition

#### 3.3.1. Photosynthesis

Crops’ capacity for photosynthetic growth can be enhanced by using nano-fertilizers. The chloroplast and other cellular components may interact with the nano fertilizer once it has entered the cytoplasm. The physiological and biochemical indexes of crops are significantly improved, and the performance of crops is improved, thus improving the nutritional quality. Plant essential nutrients Mg [81,82,83], Fe [83,84], Ca [85], and other elements of plant photosynthesis play a role, which can affect the chlorophyll content, redox reaction and thus affect plant protein content, and photosynthesis, affect the plant’s organic nutritional quality (Table 2). Studies have shown that mesoporous silica nanoparticles can enhance crop photosynthesis [86]. n-TiO_2_ can improve the photosynthetic pigment content and photosynthetic efficiency of tomatoes [87], corn [88], barley [89], and spinach [90], and can also improve the biomass and productivity of corn [89,91]. Biocompatible magnetic nanofluids can enhance the chlorophyll content of sunflower [92]. Nano Zn can improve the antioxidant capacity and cellular antioxidant activity of rice [93,94], it can also increase the biomass of sunflower [95], improve corn yield under drought conditions [96], wheat grain yield under stress [97], improve nitrogen utilization efficiency. Furthermore, it has been demonstrated that the crop yield and nutritional value of agricultural goods can be increased by using nano-chelated Mo, n-Fe, n-Mn, and n-Zn [98,99,100,101].

NF affects photosynthesis favorably and negatively. Changing chlorophyll content, electron transport rate, and chloroplast absorption spectrum will affect plant photosynthesis, and plant photosynthesis is closely related to its organic nutrition. Only by deeply understanding the interaction between NF and chloroplasts and their structures can we make better use of NF [102].

**Table 2 nanomaterials-12-04219-t002:** Nanomaterials as nutrients and their sources enhance plant photosynthesis.

Crop and Conditions	Size and Frequency	Comments	References
Macrotyloma uniflorum (*horse gram*).	MgO-NPs (10, 50, 100, and 150 μg/mL)	When exposed to MgO-NP, plant chlorophyll content increased significantly, and the accumulation of carbohydrates and protein increased by 4–20% and 18–127%, respectively.	[81]
*Black-Eyed Pea*. Magnesium nano-fertilizers. Foliar application.	Fe (0, 0.25, and 0.5 g/L); Mg (0, 0.5 g/L) nano, and common	0.5 g/L nano- iron improves chlorophyll content more efficiently than ordinary Fe.	[83]
Tomato. soil construction	(Conventional, FeCl_3_.6H_2_O; Chelated with 6% Fe; Nano Fe_2_O_3_, 99%, 30–50 nm) and Fe (0, 50, and 100 mg/kg soil)	100 mg/kg Nano-Fe can significantly improve the efficiency of tomato photosynthesis.	[84]
Barley (*Hordeum vulgare L.*). Hydroponically	Nano-HF (Sr_0.96_Mg_0.02_Ca_0.02_Fe_12_O_19_ nano-hexaferrite) (125, 250, 500, and 1000 mg/L)	When treated with 125 or 250 mg/L nano-HF, the growth effect was the strongest. Nano-HF raised the number of soluble proteins by about 41% and the amount of chlorophyll pigment by around 33–22% when compared to the untreated control.	[85]
Tomato. Mild Heat Stress.	Nano-TiO_2_.	When minor heat stress is present, tomato plants’ ability to photosynthesize can be significantly improved by adding the right amount of nano titanium dioxide.	[87]

#### 3.3.2. Hormone and Enzyme Activity

The growth conditions of plants are usually constantly changing, and plants usually adapt to adverse environmental conditions through the regulation of plant hormones [103]. Hormone imbalance will impact plant development and metabolism, making it impossible to ensure the nutritious value of crops. Nano fertilizer can stimulate the production of enzymes, enhance the activity of enzymes, help improve the tolerance of plants to various abiotic stresses, enable crops to grow and develop better, reduce the impact of adverse growth conditions, and ensure their nutritional quality [104]. The cell membrane, DNA, proteins, and other cell processes could be harmed by active oxygen. The metabolism of crops is the primary cause. The use of nano fertilizer containing antioxidant enzymes can reduce the damage of active oxygen and enhance the tolerance of crops to environmental stress [104,105,106]. The research shows that TiO_2_ nano fertilizer can regulate the production of enzymes, show the protection of enzyme and non-enzyme stress, promote the collection of nutrients and production, and improve the nutritional quality of its crops [107]. By promoting antioxidant enzymes, silica nano fertilizer can enhance seedling growth as well as their tolerance to and resilience to abiotic stress [108,109]. Cerium oxide nanoparticles (CeO_2_ NPs) can alleviate nitrogen stress in rice [110]. Cu can be used as an integral part of the enzyme, and can also be used as a cofactor for most enzymes. The experimental investigation of corn solution culture and copper nanoparticle spray application revealed that Cu nanoparticles accumulated in organisms and aided in the growth of maize and that their presence also had a substantial impact on the activity of G-6-PD [111]. APX, CAT, and GR activities were increased in the research of cowpea subjected to nano-copper, but SOD activity was lowered in the leaves and roots [75].

#### 3.3.3. Combined Application of Different Types of Fertilizers

Compared with traditional fertilizers, nano-fertilizers have obvious advantages combined fertilization is often used for different problems of plants or different requirements for their growth and nutrition. The combined application of some fertilizers will produce antagonistic effects and reduce the fertilizer efficiency of each other, and the appropriate combination can amplify the efficacy of fertilizers and meet various needs. The study of Azimi et al. showed that seed priming with gibberellin and foliar spraying of Zn + Fe increased plant height by 13%. The grain yield was increased by 18% by introducing nano-zinc and spraying iron and zinc. Nano zinc primer sprayed with iron + zinc foliar spray increased grain protein by 21% and significantly increased chlorophyll content in leaves [112]. With the help of additional irrigation and a combination foliar spray of B + Mn + Zn nano chelate, the greatest wheat yield of 2528.33 kg/ha was produced [113]. Foliar applications of Fe, Zn, and Mg nano-chelate may boost plant production, plant dry weight, and chlorophyll concentration while assisting soybean to withstand drought stress [114]. The urea-modified hydroxyapatite nano-mixed fertilizer was synthesized by the weight ratio of urea to hydroxyapatite 6:1, and the farmland experiment was carried out on rice. The outcomes demonstrated that the nano-hybrid may accelerate plant development, boost output, and reduce 50% N while ensuring plant uptake, significantly increasing fertilizer use rate [115]. Fe_3_O_4_-urea nanocomposites with different urea ratios were used to hydroponically cultivate *Oryza sativa L. cv. Swarna* plants. The photosynthetic rate and grain nutrient content increased. With the continuous release of urea, the expression profiles of ammonia and nitrate transporters changed significantly, and the nitrogen utilization efficiency increased [116].

In general, nano fertilizer has incomparable advantages over traditional fertilizer. It transports nutrients in plants in the form of nanoparticles, realizes directional and quantitative transport of nutrients, and reduces loss and waste. At the same time, nano fertilizer can also improve crop yield by promoting photosynthesis, stimulating hormones and regulating enzymes, and enhancing crop tolerance to stress. Nano fertilizer has made great contributions to promoting agricultural technology and ameliorating the nutritional quality of crops.

### 3.4. Nano Pesticide (NP)

The increasing demand for grain production poses a great challenge to traditional agriculture, which is heavily dependent on pesticides. So far, the global use of pesticides has exceeded 4.1 million tons [117], and less than 1% of the applied pesticides can achieve the purpose. Most of the remaining pesticides enter the soil, water, and atmospheric environment through degradation, volatilization, and photolysis, causing serious pollution to the environment [118,119]. As a result, modern pesticide needs include controlling release behavior, maintaining chemical stability, and having the ability to target specific areas. Nanotechnology can play a significant role in these elements [120]. Nanomaterials can replace organic solvents in traditional pesticides, and reduce environmental pollution and health risks while meeting current needs [121]. The most attractive feature is that nano pesticides can be controlled to release [122]. According to the rate and mode of regulated release, nano pesticides can be divided into slow-release nano-pesticides and stimulated-release nano-pesticides [123].

#### 3.4.1. Slow-Release Nano Pesticide

Slow-release nano pesticides are transported and released through nanocarriers. According to the material and structure of the nanocarrier, the pesticide is encapsulated or adsorbed on the nanocarrier for slow release through passive diffusion, capsule erosion, or penetration-driven penetration [123]. Memarizadeh et al. confirmed that nano-indole kappa has a larger loading capacity and lower release rate, and can reduce the quantity of pesticide necessary for pest management by successfully creating nano-indoxacarb (IND) with photo degradability and biocompatibility through supramolecular interaction [124]. Nano silica particles can be used to make sustained-release nano-pesticides by changing their structure and shape. For instance, Pohl et al. created a new delivery system for post-loading the biopesticide citric alcohol into mesoporous silica nanoparticles (MESINP) functionalized by nitrogen-rich derived polymers, proving that this nano-system can double as a soil conditioner as well as a nanocarrier for controlling and gradually releasing pesticides [125,126]. Additionally, hollow porous silica nanospheres (BHSNs) with a bowl-like structure were added to the model pesticide imidacloprid, which can aid to create the pesticide’s prolonged release through an adsorption and retention mechanism between the pesticide and the carrier [127,128]. Su et al. created novel nano-pesticides with high biological activity and slow-release properties that can control the trunk borers by self-assembling the hydrophobic pesticide thicloline (THI) inside of the oleophobic feature of BSA [129]. According to studies, carbon nanomaterials (CNMs) can be employed as a vehicle for nanoscale sustained-release insecticides [130].

#### 3.4.2. Stimulate the Release of Nano Pesticides

By modifying environmental variables including temperature, light, and pH, the stimulated release of nano insecticides can achieve the targeted and intelligent release of biological or non-biological stimuli [123]. Stimulated release of nano pesticides can generally be divided into two categories: valve-controlled release of nano pesticides and overall stimulated release of nano pesticides. Valve-controlled release of nano pesticides is mainly realized by designing a valve sealing layer and using the inorganic core carrier. In recent work, the pesticide chlorpyrifos benzamide was deposited onto mesoporous silica, and a valve layer was created on its surface. When it enters the insect body, the enzyme in the insect gut will hydrolyze the valve layer and release the pesticide to kill the insect [131]. A light-responsive valve can also be designed to control the release of nano pesticides under the stimulation of ultraviolet-visible light [132] and infrared light [133]. In addition, the release of nano pesticides can be stimulated under temperature changes or high and low temperatures through temperature control valve layers [134,135].

With the wide application of nanotechnology, scholars continue to make efforts in the research of controlling pesticide release behavior, improving targeting ability, reducing pesticide application amount, and improving use efficiency by using nanotechnology. By using the special structure and properties of nanomaterials, the release of pesticides can be controlled by adsorption and encapsulation (Figure 2). Crops can be protected from pests, and, to a certain extent, the nutritious value of agricultural products can be ensured by the precise and controllable release of pesticides.

### 3.5. Plant Toxicity of Nanomaterials

Plants growing in soils can have adverse effects even when exposed to low concentrations of nanomaterials, mainly due to the poor mobility of soils, where nanomaterials tend to accumulate [91,136,137,138]. In the whole cycle of crop growth, according to previous studies, we found that nanomaterials have produced toxicity in different periods of plants. The germination rates of zucchini seeds exposed to sodium dodecyl sulfate (SDS) modified Ag-NMs, multi-walled carbon nanotubes (MWCNTs) and Si-NMs decreased by 54%, 47%, and 80%, respectively [139]. Similarly, Canas et al. compared the effects of functionalized and non-functionalized nanotubes on the early growth of crops. They found that functionalized nanotubes inhibited the root elongation of lettuce, and non-functionalized nanotubes inhibited the root elongation of tomato [140]. Ce_2_O-NMs are widely used in the study of crop drought stress, but at the same time, we cannot ignore their possible toxicity. Studies have found that soybean exposure to Ce_2_O-NPs (8 nm, 0.5 g/kg soil) can cause soybean ROS and lipid peroxidation to increase and total chlorophyll concentration to decrease [141]. In addition, Ce_2_O-NMs (8 ± 1 nm, 100 and 400 mg/kg) were also found to delay the flowering period of wheat by one week, which would lead to a shortened filling period of wheat [142]. At the same time, Ke et al. revealed that Ag-NMs (12.5 mg/kg soil) could delay the flowering of Arabidopsis by 5 days, which may be related to the down-regulation of AP1, SOC1, FT, and LFY genes in Arabidopsis [143]. At the maturity stage of crops, we also found the toxicity of NMs. For example, soybeans exposed to ZnO-NMs (0.5/kg soil) did not form seeds during the final harvest period [144]. Studies have shown that changing the application concentration, exposure time and other conditions can alleviate the plant toxicity of nanomaterials [145]. However, at present, the research on the mechanism of plant toxicity of nanomaterials is not clear enough. In order to realize the comprehensive promotion and use of nanomaterials, the toxicity of nanomaterials needs to be further studied.

## 4. Summary and Prospects

Nanotechnology has been extensively utilized in a number of industries and aspects of daily life in the past. Nanomaterials’ unique structure offers the potential for green and sustainable development as well as new development options for established industries. Particularly in agriculture, nanotechnology has nearly made an original contribution. Agrochemicals like nano-pesticides, nano-fertilizers, and nano-sensors have continuously increased the yield and nutritional value of agricultural products increased the efficiency with which pesticides and fertilizers are used, and decreased the environmental pollution brought on by the loss of agrochemicals. The potential for scaling up nanotechnology’s use in agriculture is limitless. However, at the same time, nano agrochemicals have not been put into use on a large scale, because the toxicity of nanomaterials and the harm caused by environmental residues have not been thoroughly understood and solved. The emergence and application of new technologies will inevitably encounter many difficulties. While seizing the opportunities, we must also meet the challenges. Nanotechnology needs to meet the requirements of specific plant types, soil conditions, climate conditions, growth needs, and other conditions and can be used on a large scale only after its toxicity and threat to human health are completely solved. Here are some expectations for future research directions:The current research on the toxicity of nanomaterials still needs to be deepened. In the future, we can focus on determining the causes and influencing factors of toxicity and finding solutions.The future needs a better understanding of nanomaterials and plants, and soil interaction mechanisms, through the data, to see the reasons behind and mechanism.The related research on the combined application of nano-fertilizers and traditional fertilizers needs to be further studied to improve economic benefits while ensuring fertilizer efficiency.At present, there are few studies on the treatment of nanocarriers of nanomaterial-loaded fertilizers. Future research can focus on solving the possible toxicity problems caused by carrier residues in plants or soils and exploring methods for their degradation or reuse.

## Figures and Tables

**Figure 1 nanomaterials-12-04219-f001:**
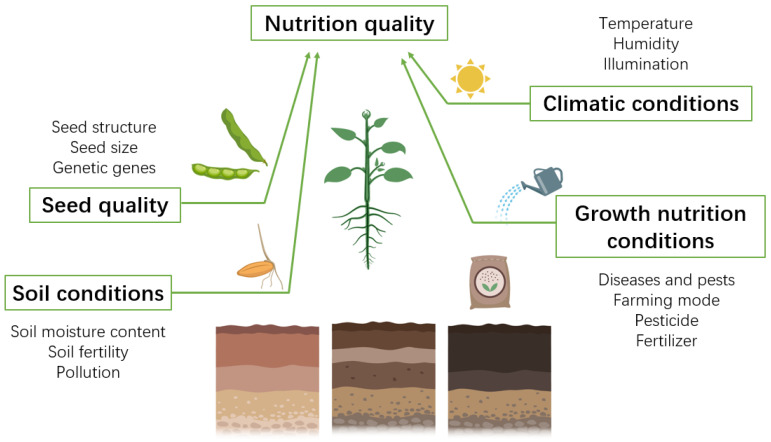
Influencing factors of crop nutrition quality.

**Figure 2 nanomaterials-12-04219-f002:**
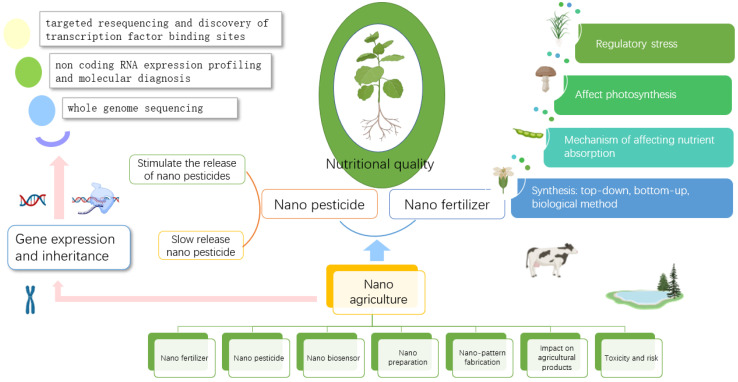
Schematic diagram of nano-agriculture, mainly to improve the nutritional quality of agricultural products by improving crop genes, nano pesticides, and nano fertilizers.

## Data Availability

Not applicable.

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
