# Peer review of "Engineered Nanomaterials for Improving the Nutritional Quality of Agricultural Products: A Review"

_nanomaterials, 2022, doi:10.3390/nano12234219_

Round 1

Reviewer 1 Report

Journal: Nanomaterials

Ms. ID.: nanomaterials-2038670

Title: Engineered nanomaterials for improving the nutritional quality of agricultural products: A review

Sun et al. tried to review the research progress of using nanomaterials to improve nutritional quality in recent years and point out the focus of future research. Unfortunately, the manuscript is predominantly inconclusive and confusing. The introduction should be more focused and straight-forward. The motivation for this manuscript is not clear. The use of nanomaterials in agriculture is not obvious in this manuscript. It should be completely rewritten in order to be published.

Some specific comments:

-Sections 2.1. and 2.2. have the same title.

-Figure 1 needs to be improved.

-Line 212: “Nanoparticles or nanomaterials are made into nano fertilizers.” What does it mean?

-Section 3.1. is completely confusing.

-Section 3.4.3. Combined use of what?

- Line 489: What are NMs?

-Section 5. deserves more attention.

I suggest the authors rewrite the manuscript and submit it again since this topic is very interesting.

Author Response

Thanks for the suggestion. Please see the attachment.

Reviewer 2 Report

The manuscript by Sun et al. is an interesting review, however, it has some mistakes and corrections needs to be made before its final publication.

Comments:

L54: Sustainable.

L59: Italicize Z. japonica

L70: Nano-pesticides x2?

L48 and L77-79: Since nanomaterials and in particular nanoparticles have shown significant toxicities to various agricultural crops even at lower test concentrations, therefore, a section addressing this aspect should also be discussed in brief.

A few papers are suggested here which may help authors to write a brief section on nano-toxicity to plants:

Chemosphere 300, 134555

Environmental Pollution 289, 117854

https://doi.org/10.1016/B978-0-323-90774-3.00013-1

Journal of Hazardous Materials 419, 126493

Environmental Pollution 310, 119916

Environmental Chemistry Letters 19 (2), 1545-1609

Chemosphere 281, 130940

Environmental Pollution 240, 802-816

Please check sections 2.1 and 2.2, are they same?

Section 3.1: There is less discussion about the synthesis and mechanism of synthesis of nano-fertilizer but more is about type of nano-fertilizer. also, authors need to figure out how nanoparticles are different from nanofertilizers? 

Authors need to check the Table 1 very carefully. Italicize all scientific names.

Calcium carbonate nanoparticles. 

Thanks.   ??

There are several other typos and word mistakes.

L373: What is NPS?

Caption of Table 2 is wrong.

Also check all the scientific names.

Figure 2: diagram of agriculture or nano-agriculture? 

In the whole manuscript, nowhere I found any commercially available or FDA approved nanopesticides or nanofertilizers. Can authors put some efforts to include a few of approved nano-based pesticides and/fertilizers, possibly in a separate section or table with their mechanism of action and benefits over non-nano pesticides and fertilizers?

Try to shorten the bullet points in prospects.

Author Response

(The authors gave the same response as above.)

Round 2

Reviewer 1 Report

The authors made an effort to improve the manuscript, but there are still more improvements to be made, in my opinion. The structure of the manuscript is the same as before. The introduction is slightly improved, but still not satisfying. The motivation for this manuscript and its novelty should be claimed within the introduction. Specific comments were addressed well.

Round 3

Reviewer 1 Report

The authors addressed all my comments.